# Automata Guided Hierarchical Reinforcement Learning for Zero-Shot Skill Composition

## Abstract

An obstacle that prevents the wide adoption of (deep) reinforcement learning (RL) in control systems is its need for a large number of interactions with the environment in order to master a skill. The learned skill usually generalizes poorly across domains and re-training is often necessary when presented with a new task. We present a framework that combines techniques in *formal methods* with *hierarchical reinforcement learning* (HRL). The set of techniques we provide allows for the convenient specification of tasks with logical expressions, learns hierarchical policies (meta-controller and low-level controllers) with well-defined intrinsic rewards using any RL methods and is able to construct new skills from existing ones without additional learning. We evaluate the proposed methods in a simple grid world simulation as well as simulation on a Baxter robot.

## 1 Introduction

Reinforcement learning has received much attention in the recent years because of its achievements in games Mnih et al. (2015), Silver et al. (2016), robotics manipulation Jang et al., Levine et al. (2016), Gu et al. (2016) and autonomous driving Isele et al. (2017), Madrigal (2017). However, training a policy that sufficiently masters a skill requires an enormous amount of interactions with the environment and acquiring such experience can be difficult on physical systems. Moreover, most learned policies are tailored to mastering one skill (by maximizing the reward) and are hardly reusable on a new skill.

Skill composition is the idea of constructing new skills out of existing skills (and hence their policies) with little to no additional learning. In stochastic optimal control, this idea has been adopted by authors of Todorov (2009) and Da Silva et al. (2009) to construct provably optimal control laws based on linearly solvable Markov decision processes. Authors of Haarnoja et al. (2017), Tang & Haarnoja have showed in simulated manipulation tasks that approximately optimal policies can result from adding the Q-functions of the existing policies.

Hierarchical reinforcement learning is an effective means of achieving transfer among tasks. The goal is to obtain task-invariant low-level policies, and by re-training the meta-policy that schedules over the low-level policies, different skills can be obtain with less samples than training from scratch. Authors of Heess et al. (2016) have adopted this idea in learning locomotor controllers and have shown successful transfer among simulated locomotion tasks. Authors of Oh et al. (2017) have utilized a deep hierarchical architecture for multi-task learning using natural language instructions.

Temporal logic is a formal language commonly used in software and digital circuit verification Baier & Katoen (2008) as well as formal synthesis Belta et al. (2017). It allows for convenient expression of complex behaviors and causal relationships. TL has been used by Sadraddini & Belta (2015), Leahy et al. (2015) to synthesize provably correct control policies. Authors of Aksaray et al. (2016) have also combined TL with Q-learning to learn satisfiable policies in discrete state and action spaces.

In this work, we focus on hierarchical skill acquisition and zero-shot skill composition. Once a set of skills is acquired, we provide a technique that can synthesize new skills without the need to further interact with the environment (given the state and action spaces as well as the transition remain the

same). We adopt temporal logic as the task specification language. Compared to most heuristic reward structures used in the RL literature to specify tasks, formal specification language excels at its semantic rigor and interpretability of specified behaviors. Our main contributions are:

- We take advantage of the transformation between TL formula and finite state automata (FSA) to construct deterministic meta-controllers directly from the task specification without the necessity for additional learning. We show that by adding one discrete dimension to the original state space, structurally simple parameterized policies such as feed-forward neural networks can be used to learn tasks that require complex temporal reasoning.

- Intrinsic motivation has been shown to help RL agents learn complicated behaviors with less interactions with the environment Singh et al. (2004), Kulkarni et al. (2016), Jaderberg et al. (2016). However, designing a well-behaved intrinsic reward that aligns with the extrinsic reward takes effort and experience. In our work, we construct intrinsic rewards directly from the input alphabets of the FSA (a component of the automaton), which guarantees that maximizing each intrinsic reward makes positive progress towards satisfying the entire task specification. From a user's perspective, the intrinsic rewards are constructed automatically from the TL formula.

- In our framework, each FSA represents a hierarchical policy with low-level controllers that can be re-modulated to achieve different tasks. Skill composition is achieved by manipulating the FSA that results from their TL specifications in a deterministic fashion. Instead of interpolating/extrapolating among existing skills, we present a simple policy switching scheme based on graph manipulation of the FSA. Therefore, the compositional outcome is much more transparent. We introduce a method that allows learning of such hierarchical policies with any non-hierarchical RL algorithm. Compared with previous work on skill composition, we impose no constraints on the policy representation or the problem class.

## 2 PRELIMINARIES

### 2.1 THE OPTIONS FRAMEWORK IN HIERARCHICAL REINFORCEMENT LEARNING

In this section, we briefly introduce the options framework Sutton et al. (1998), especially the terminologies that we will inherit in later sections. We start with the definition of a Markov Decision Process.

**Definition 1.** *An MDP is defined as a tuple $\mathcal{M} = \langle S, A, p(\cdot|\cdot, \cdot), R(\cdot, \cdot, \cdot) \rangle$, where $S \subseteq \mathbb{R}^n$ is the state space ; $A \subseteq \mathbb{R}^m$ is the action space ($S$ and $A$ can also be discrete sets); $p : S \times A \times S \to [0, 1]$ is the transition function with $p(s'|s, a)$ being the conditional probability density of taking action $a \in A$ at state $s \in S$ and ending up in state $s' \in S$; $R : S \times A \times S \to \mathbb{R}$ is the reward function. let $T$ be the length of a fixed time horizon. The goal is to find a policy $\pi^\star : S \to A$ (or $\pi^\star : S \times A \to [0, 1]$ for stochastic policies) that maximizes the expected return, i.e.*

$$\pi^\star = \arg\max_\pi \mathbb{E}^\pi[R(\tau_T)] \tag{1}$$

*where $\tau_T = (s_0, a_0, ..., s_T, )$ denotes the state-action trajectory from time $0$ to $T$.*

The options framework exploits temporal abstractions over the action space. An option is defined as a tuple $o = \langle \mathcal{I}, \pi^o, \beta \rangle$ where $\mathcal{I}$ is the set of states that option $o$ can be initiated (here we let $\mathcal{I} = S$ for all options), $\pi^o : S \to A$ is an options policy and $\beta : S \to [0, 1]$ is the termination probability for the option at state $s$. In addition, there is a policy over options $\pi^h : S \to O$ (where $O$ is a set of available options) that schedules among options. At a given time step $t$, an option $o$ is chosen according to $\pi^h(s_t)$ and the options policy $\pi^o$ is followed until the termination probability $\beta(s) > threshold$ at time $t + k$, and the next option is chosen by $\pi^h(s_{t+k})$.

## 2.2 scTLTL and Automata

We consider tasks specified with *Truncated Linear Temporal Logic* (TLTL). We restrict the set of allowed operators to be

$$\phi := \top \mid f(s) < c \mid \neg\phi \mid \phi \wedge \psi \mid \phi \vee \psi \mid$$
$$\Diamond\phi \mid \phi\,\mathcal{U}\,\psi \mid \phi\,\mathcal{T}\,\psi \mid \bigcirc\phi \mid \phi \Rightarrow \psi \tag{2}$$

where $f(s) < c$ is a predicate, $\neg$ (negation/not), $\wedge$ (conjunction/and), and $\vee$ (disjunction/or) are Boolean connectives, and $\Diamond$ (eventually), $\mathcal{U}$ (until), $\mathcal{T}$ (then), $\bigcirc$ (next), are temporal operators. Implication is denoted by $\Rightarrow$ (implication). Essentially we excluded the *Always* operator ($\Box$) with reasons similar to Kupferman & Vardi (2001). We refer to this restricted TLTL as *syntactically co-safe TLTL* (scTLTL) (Vasile et al. (2017) used similar idea for LTL). There exists a real-value function $\rho(s_{0:T}, \phi)$ called robustness degree that measures the level of satisfaction of trajectory $s_{0:T}$ with respective to $\phi$. $\rho(s_{0:T}, \phi) > 0$ indicates that $s_{0:T}$ satisfies $\phi$ and vice versa. Definitions for the boolean semantics and robustness degree are provided in Appendix E.

Any scTLTL formula can be translated into a finite state automata (FSA) with the following definition:

**Definition 2.** *An FSA is defined as a tuple* $\mathcal{A}_\phi = \langle Q_\phi, \Psi_\phi, q^0, p_\phi(\cdot|\cdot), \mathcal{F}_\phi \rangle$, *where* $Q_\phi$ *is a set of automaton states;* $\Psi_\phi$ *is an input alphabet, we denote* $\psi_{q_i,q_j} \in \Psi_\phi$ *the predicate guarding the transition from* $q_i$ *to* $q_j$ *(as illustrated in Figure 1 );* $q^0 \in Q_\phi$ *is the initial state;* $p_\phi : Q_\phi \times Q_\phi \to [0,1]$ *is a conditional probability defined as*

$$p_\phi(q_j|q_i) = \begin{cases} 1 & \psi_{q_i,q_j} \text{ is true} \\ 0 & \text{otherwise.} \end{cases} \tag{3}$$

*In addition, given an MDP state* $s$, *we can calculate the transition in automata states at* $s$ *by*

$$p_\phi(q_j|q_i, s) = \begin{cases} 1 & \rho(s, \psi_{q_i,q_j}) > 0 \\ 0 & \text{otherwise.} \end{cases} \tag{4}$$

*We abuse the notation* $p_\phi$ *to represent both kinds of transitions when the context is clear.* $\mathcal{F}_\phi$ *is a set of final automaton states.*

The translation from TLTL formula to FSA to can be done automatically with available packages like Lomap Ulusoy (2017).

**Example 1.** *Figure 1 (left) illustrates the FSA resulting from formula* $\phi = \neg b\,\mathcal{U}\,a$. *In English,* $\phi$ *entails during a run,* $b$ *cannot be true until* $a$ *is true and* $a$ *needs to be true at least once. The FSA has three automaton states* $Q_\phi = \{q_0, q_f, trap\}$ *with* $q_0$ *being the input(initial) state (here* $q_i$ *serves to track the progress in satisfying* $\phi$). *The input alphabet is defined as the* $\Psi_\phi = \{\neg a \wedge \neg b, \neg a \wedge b, a \wedge \neg b, a \wedge b\}$. *Shorthands are used in the figure, for example* $a = (a \wedge b) \vee (a \wedge \neg b)$. $\Psi_\phi$ *represents the power set of* $\{a, b\}$, *i.e.* $\Psi_\phi = 2^{\{a,b\}}$. *During execution, the FSA always starts from state* $q_0$ *and transitions according to Equation* (3) *or* (4). *The specification is satisfied when* $q_f$ *is reached and violated when* $trap$ *is reached. In this example,* $q_f$ *is reached only when* $a$ *becomes true before* $b$ *becomes true.*

## 3 Problem Formulation and Approach

We start with the following problem definition:

**Problem 1.** *Given an MDP in Definition 1 with unknown transition dynamics* $p(s'|s, a)$ *and a scTLTL specification* $\phi$ *over state predicates (along with its FSA* $\mathcal{A}_\phi$) *as in Definition 2. Find a policy* $\pi_\phi^\star$ *such that*

$$\pi_\phi^\star = \arg\max_{\pi_\phi} \mathbb{E}^{\pi_\phi}[\mathbb{1}(\rho(s_{0:T}, \phi) > 0)]. \tag{5}$$

*where* $\mathbb{1}(\rho(s_{0:T}, \phi) > 0)$ *is an indicator function with value 1 if* $\rho(s_{0:T}, \phi) > 0$ *and 0 otherwise.*

Problem 1 defines a policy search problem where the trajectories resulting from following the optimal policy should satisfy the given scTLTL formula in expectation.

**Problem 2.** *Given two scTLTL formula $\phi_1$ and $\phi_2$ along with policy $\pi_{\phi_1}$ that satisfies $\phi_1$ and $\pi_{\phi_2}$ that satisfies $\phi_2$. Obtain a policy $\pi_\phi$ that satisfies $\phi = \phi_1 \wedge \phi_2$.*

Problem 2 defines the problem of task composition. Given two policies each satisfying a scTLTL specification, construct the policy that satisfies the conjunction of the given specifications. Solving this problem is useful when we want to break a complex task into simple and manageable components, learn a policy that satisfies each component and "stitch" all the components together so that the original task is satisfied. It can also be the case that as the scope of the task grows with time, the original task specification is amended with new items. Instead of having to re-learn the task from scratch, we can only learn a policy that satisfies the new items and combine them with the old policy.

We propose to solve Problem 1 by constructing a product MDP from the given MDP and FSA that can be solved using any state-of-the-art RL algorithm. The idea of using product automaton for control synthesis has been adopted in various literature Leahy et al. (2015), Chen et al. (2012). However, the methods proposed in these works are restricted to discrete state and actions spaces. We extend this idea to continuous state-action spaces and show its applicability on robotics systems.

For Problem 2, we propose a policy switching scheme that satisfies the compositional task specification. The switching policy takes advantage of the characteristics of FSA and uses robustness comparison at each step for decision making.

## 4 FSA AUGMENTED MDP

Problem 1 can be solved with any episode-based RL algorithm. However, doing so the agent suffers from sparse feedback because a reward signal can only be obtained at the end of each episode. To address this problem as well as setting up ground for automata guided HRL, we introduce the FSA augmented MDP

**Definition 3.** *An FSA augmented MDP corresponding to scTLTL formula $\phi$ is defined as $\mathcal{M}_\phi = \langle \tilde{S}, A, \tilde{p}(\cdot|\cdot, \cdot), \tilde{R}(\cdot, \cdot) \rangle$ where $\tilde{S} \subseteq S \times Q_\phi$, $A$ is the same as the original MDP. $\tilde{p}(\tilde{s}'|\tilde{s}, a)$ is the probability of transitioning to $\tilde{s}'$ given $\tilde{s}$ and $a$, in particular*

$$\tilde{p}(\tilde{s}'|\tilde{s}, a) = p\big((s', q')|(s, q), a\big)$$
$$= \begin{cases} p(s'|s, a) & p_\phi(q'|q, s) = 1 \\ 0 & otherwise. \end{cases} \tag{6}$$

*Here $p_\phi$ is defined in Equation (4). $\tilde{R} : \tilde{S} \times \tilde{S} \to \mathbb{R}$ is the FSA augmented reward function, defined by*

$$\tilde{R}(\tilde{s}, \tilde{s}') = \mathbb{1}\big(\rho(s', D_\phi^q) > 0\big)^1 \tag{7}$$

*where $\Omega_q$ is the set of automata states that are connected with $q$ through outgoing edges. $D_\phi^q = \bigvee_{q' \in \Omega_q} \psi_{q,q'}$ represents the disjunction of all predicates guarding the transitions that originate from $q$. The goal is to find the optimal policy that maximizes the expected sum of discounted return, i.e.*

$$\pi^\star = \arg\max_\pi \mathbb{E}^\pi \left[ \sum_{t=0}^{T-1} \gamma^{t+1} \tilde{R}(s_t, s_{t+1}) \right], \tag{8}$$

*where $\gamma < 1$ is the discount factor, $T$ is the time horizon.*

As a quick example to the notation $D_\phi^q$, consider the state $q_0$ in the FSA in Figure 1 , $\Omega_{q_0} = \{trap, q_f\}$, $D_\phi^{q_0} = \psi_{q_0, trap} \vee \psi_{q_0, qf} = b \vee a$. The goal is then to find a policy $\pi : \tilde{S} \to A$ that maximizes the expected sum of $\tilde{R}$ over the horizon $T$.

---

[1] Because $D_\phi^q$ is a predicate without temporal operators, the robustness $\rho(s_{t:t+k}, D_\phi^q)$ is only evaluated at $s_t$(refer to Appendix E). Therefore, we use the shorthand $\rho(s_t, D_\phi^q) = \rho(s_{t:t+k}, D_\phi^q)$

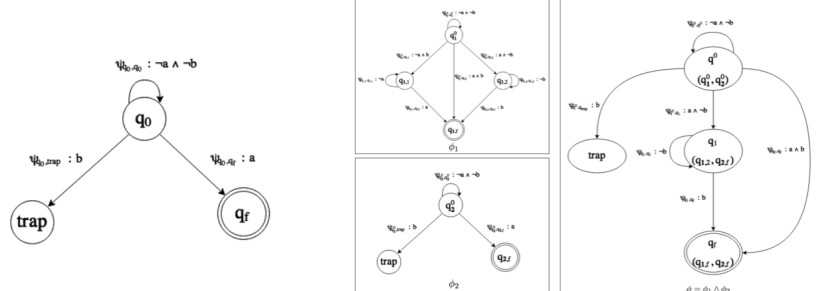

**Figure 1 :** FSA constructed from $\phi = \neg b \; \mathcal{U} \; a$. (*right*): Specification amendment example. $\mathcal{A}_{\phi_1}$ is constructed from $\phi_1 = \Diamond a \wedge \Diamond b$. $\mathcal{A}_{\phi_2}$ is constructed from $\phi_2 = \neg b \mathcal{U} a$. $\mathcal{A}_\phi$ is constructed from $\phi = \phi_1 \wedge \phi_2$. The automaton state pair in parenthesis denote the corresponding states from $Q_{\phi_1}$ and $Q_{\phi_2}$ that the product state is constructed from. $q_i^0$ denotes the initial state of $\mathcal{A}_{\phi_i}$, $q_{i,j}$ denotes the $j^{th}$ state of $Q_{\phi_i}$.

The FSA augmented MDP can be constructed with any standard MDP and a scTLTL formula. And it can be solved with any off-the-shelf RL algorithm. By directly learning the flat policy $\pi$ we bypass the need to learn multiple options policies separately. After obtaining the optimal policy $\pi^\star$, the optimal options policy for any option $o_q$ can be extracted by executing $\pi^\star(a|s,q)$ without transitioning the automata state, i.e. keeping $q_i$ fixed (denoted $\pi_q^\star$ ). And $\pi_q^\star$ satisfies

$$\pi_{q_i}^\star = \arg\max_{\pi_{q_i}} \mathbb{E}^{\pi_{q_1}} \left[ \sum_{t=0}^{T-1} \gamma^{t+1} \mathbb{1}\big(\rho(s_{t+1}, D_\phi^{q_i}) > 0\big) \right]. \tag{9}$$

In other words, the purpose of $\pi_{q_i}$ is to activate one of the outgoing edges of $q_i$ as soon as possible and by doing so repeatedly eventually reach $q_f$.

The reward function in Equation (7) encourages the system to exit the current automata state and move on to the next, and by doing so eventually reach the final state $q_f$. However, this reward does not distinguish between the $trap$ state and other states and therefore will also promote entering of the $trap$ state. One way to address this issue is to impose a terminal reward on both $q_f$ and $trap$. Because the reward is an indicator function with maximum value of 1, we assign terminal rewards $R_{q_f} = 2$ and $R_{trap} = -2$.

Appendix D describes the typical learning routine using FSA augmented MDP. The algorithm utilizes memory replay which is popular among off-policy RL methods (DQN, A3C, etc) but this is not a requirement for learning with $\tilde{M}_\phi$. On-policy methods can also be used.

## 5 AUTOMATA GUIDED TASK COMPOSITION

In section, we provide a solution for Problem 2 by constructing the FSA of $\phi$ from that of $\phi_1$ and $\phi_2$ and using $\phi$ to synthesize the policy for the combined skill. We start with the following definition.

**Definition 4.** *Given* $\mathcal{A}_{\phi_1} = \langle Q_{\phi_1}, \Psi_{\phi_1}, q_1^0, p_{\phi_1}, \mathcal{F}_{\phi_1} \rangle$ *and* $\mathcal{A}_{\phi_2} = \langle Q_{\phi_2}, \Psi_{\phi_2}, q_2^0, p_{\phi_2}, \mathcal{F}_{\phi_2} \rangle$, *The FSA of* $\phi$ *is the product automaton of* $\mathcal{A}_{\phi_1}$ *and* $\mathcal{A}_{\phi_1}$, *i.e.* $\mathcal{A}_{\phi = \phi_1 \wedge \phi_2} = \mathcal{A}_{\phi_1} \times \mathcal{A}_{\phi_2} = \langle Q_\phi, \Psi_\phi, q^0, p_\phi, \mathcal{F}_\phi \rangle$ *where* $Q_\phi \subseteq Q_{\phi_1} \times Q_{\phi_2}$ *is the set of product automaton, states,* $q^0 = (q_1^0, q_2^0)$ *is the product initial state,* $\mathcal{F} \subseteq \mathcal{F}_{\phi_1} \cap \mathcal{F}_{\phi_2}$ *is the final accepting states. Following Definition 2, for states* $q = (q_1, q_2) \in Q_\phi$ *and* $q' = (q_1', q_2') \in Q_\phi$, *the transition probability* $p_\phi$ *is defined as*

$$p_\phi(q'|q) = \begin{cases} 1 & p_{\phi_1}(q_1'|q_1)p_{\phi_2}(q_2'|q_2) = 1 \\ 0 & otherwise. \end{cases} \tag{10}$$

**Example 2.** *Figure 1 (right) illustrates the FSA of* $\mathcal{A}_{\phi_1}$ *and* $\mathcal{A}_{\phi_2}$ *and their product automaton* $\mathcal{A}_\phi$. *Here* $\phi_1 = \Diamond a \wedge \Diamond b$ *which entails that both* $a$ *and* $b$ *needs to be true at least once (order does not matter), and* $\phi_2 = \neg b \; \mathcal{U} \; a$ *which is the same as Example 1. The resultant product corresponds to the formula* $\phi = (\Diamond a \wedge \Diamond b) \wedge (\neg b \; \mathcal{U} \; a)$ *which dictates that* $a$ *and* $b$ *need to be true at least once, and*

*a needs to be true before b becomes true (an ordered visit). We can see that the* trap *state occurs in* $\mathcal{A}_{\phi_2}$ *and* $\mathcal{A}_\phi$*, this is because if b is ever true before a is true, the specification is violated and* $q_f$ *can never be reached. In the product automaton, we aggregate all state pairs with a trap state component into one trap state.*

For $q = (q_1, q_2) \in Q_\phi$, let $\Psi_q$, $\Psi_{q_1}$ and $\Psi_{q_2}$ denote the set of predicates guarding the outgoing edges of $q$, $q_1$ and $q_2$ respectively. Equation (10) entails that a transition at $q$ in the product automaton $\mathcal{A}_\phi$ exists only if corresponding transitions at $q_1$, $q_2$ exist in $\mathcal{A}_{\phi_1}$ and $\mathcal{A}_{\phi_2}$ respectively. Therefore, $\psi_{q,q'} = \psi_{q_1,q_1'} \wedge \psi_{q_2,q_2'}$, for $\psi_{q,q'} \in \Psi_q, \psi_{q_1,q_1'} \in \Psi_{q_1}, \psi_{q_2,q_2'} \in \Psi_{q_2}$ (here $q_i'$ is a state such that $p_{\phi_i}(q_i'|q_i) = 1$). Following Equation (9),

$$\pi_q^\star = \arg\max_{\pi_q} \mathbb{E}^{\pi_q} \Big[ \sum_{t=0}^{T-1} \gamma^{t+1} \mathbb{1}\big(\rho(s_{t+1}, D_\phi^q) > 0\big)\Big],$$
$$\text{where } D_\phi^q = \bigvee_{q_1', q_2'} (\psi_{q_1,q_1'} \wedge \psi_{q_2,q_2'}). \tag{11}$$

Repeatedly applying the distributive law $(\Delta \wedge \Omega_1) \vee (\Delta \wedge \Omega_2) = \Delta \wedge (\Omega_1 \vee \Omega_2)$ to the logic formula $D_\phi^q$ transforms the formula to

$$D_\phi^q = \big(\bigvee_{q_1'} \psi_{q_1,q_1'}\big) \wedge \big(\bigvee_{q_2'} \psi_{q_2,q_2'}\big) = D_{\phi_1}^{q_1} \wedge D_{\phi_2}^{q_2}. \tag{12}$$

Therefore,

$$\pi_q^\star = \arg\max_{\pi_q} \mathbb{E}^{\pi_q} \Big[ \sum_{t=0}^{T-1} \gamma^{t+1} \mathbb{1}\big(\rho(s_{t+1}, D_{\phi_1}^{q_1} \wedge D_{\phi_2}^{q_2}) > 0)\big)\Big]$$
$$= \arg\max_{\pi_q} \mathbb{E}^{\pi_q} \Big[ \sum_{t=0}^{T-1} \gamma^{t+1} \mathbb{1}\big( \min(\rho(s_{t+1}, D_{\phi_1}^{q_1}), \rho(s_{t+1}, D_{\phi_2}^{q_2})) > 0)\big)\Big] \tag{13}$$

The second step in Equation (13) follows the robustness definition. Recall that the optimal options policies for $q_1$ and $q_2$ satisfy

$$\pi_{q_i}^\star = \arg\max_{\pi_{q_i}} \mathbb{E}^{\pi_{\phi_i}} \Big[ \sum_{t=0}^{T-1} \gamma^{t+1} \mathbb{1}\big(\rho(s_{t+1}, D_{\phi_i}^{q_i}) > 0)\big)\Big], \ i = 1, 2. \tag{14}$$

Equation (13) provides a relationship among $\pi_q^\star$, $\pi_{q_1}^\star$ and $\pi_{q_2}^\star$. Given this relationship, We propose a simple switching policy based on stepwise robustness comparison that satisfies $\phi = \phi_1 \wedge \phi_2$ as follows

$$\pi_\phi(s, q) = \begin{cases} \pi_{\phi_1}(s, q_1) & \rho(s_t, D_{\phi_1}^{q_1}) < \rho(s_t, D_{\phi_2}^{q_2}) \\ \pi_{\phi_2}(s, q_2) & otherwise \end{cases} \tag{15}$$

We show empirically the use of this switching policy for skill composition and discuss its limitations in the following sections.

## 6 EXPERIMENTS AND DISCUSSION

### 6.1 GRID WORLD SIMULATION

In this section, we provide a simple grid world navigation example to illustrate the techniques presented in Sections 4 and 5. Here we have a robot navigating in a discrete 1 dimensional space. Its

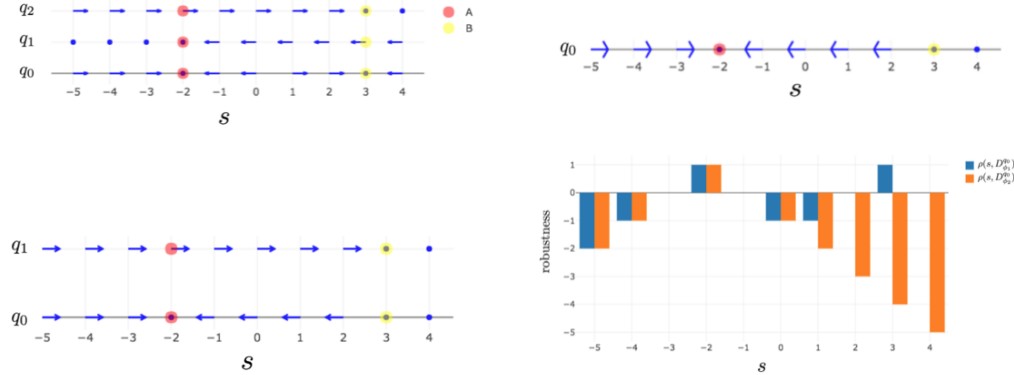

**Figure 2 :** *upper left*: Optimal policy for $\phi_1 = \Diamond a \wedge \Diamond b$ trained using Q-Learning. The arrows represent the action at each state and the dot represents stay still at that state. (*upper right*): Optimal policy for $\phi_2 = \neg b \; \mathcal{U} \; a$. (*lower left*):Optimal policy for $\phi = (\Diamond a \wedge \Diamond b) \wedge (\neg b \; \mathcal{U} \; a)$. (*lower right* ): Robustness comparison used for construction of policy $\pi^\star_{\phi_1 \wedge \phi_2}$. The robustness value is zero for states where bars disappear.

MDP state space $S = \{s | s \in [-5, 5), s \text{ is discrete}\}$, its action space $A = \{left, stay, right\}$. The robot navigates in the commanded direction with probability 0.8, and with probability 0.2 it randomly chooses to go in the opposite direction or stay in the same place. The robot stays in the same place if the action leads it to go out of bounds.

We define two regions $a : -3 < s < -1$ and $b : 2 < s < 4$. For the first task, the scTLTL specification $\phi_1 = \Diamond \; a \wedge \Diamond b$ needs to be satisfied. In English, $\phi_1$ entails that the robot needs to visit regions $a$ and $b$ at least once. To learn a deterministic optimal policy $\pi^\star_{\phi_1} : S \times Q \to A$, we use standard Q-Learning Watkins (1989) on the FSA augmented MDP for this problem. We used a learning rate of 0.1, a discount factor of 0.99, epsilon-greedy exploration strategy with $\epsilon$ decaying linearly from 0.0 to 0.01 in 1500 steps. The episode horizon is $T = 50$ and trained for 500 iterations. All Q-values are initialized to zero. The resultant optimal policy is illustrated in Figure 2 .

We can observe from the figure above that the policy on each automaton state $q$ serves a specific purpose. $\pi^\star_{q_0}$ tries to reach region $a$ or $b$ depending on which is closer. $\pi^\star_{q_1}$ always proceeds to region $a$. $\pi^\star_{q_2}$ always proceeds to region $b$. This agrees with the definition in Equation 9. The robot can start anywhere on the $s$ axis but must always start at automata state $q_0$. Following $\pi_{\phi_1}$, the robot will first reach region $a$ or $b$ (whichever is nearer), and then aim for the other region which in turn satisfies $\phi$. The states that have $stay$ as their action are either goal regions (states $(-2, q_0), (3, q_1)$, etc) where a transition on $q$ happens or states that are never reached (states $(-3, q_1), (-4, q_2)$, etc) because a transition on $q$ occurs before they can be reached.

To illustrate automata guided task composition described in Example 2, instead of learning the task described by $\phi$ from scratch, we can simply learn policy $\pi_{\phi_2}$ for the added requirement $\phi_2 = \neg b \; \mathcal{U} \; a$. We use the same learning setup and the resultant optimal policy is depicted in Figure 4 . It can be observed that $\pi_{\phi_2}$ tries to reach $a$ while avoiding $b$. This behavior agrees with the specification $\phi_2$ and its FSA provided in Figure 2 . The action at $s = 4$ is $stay$ because in order for the robot to reach $a$ it has to pass through $b$, therefore it prefers to obtain a low reward over violating the task.

Having learned policies $\pi_{\phi_1}$ and $\pi_{\phi_2}$, we can now use Equation 15 to construct policy $\pi_{\phi_1 \wedge \phi_2}$. The resulting policy for $\pi_{\phi_1 \wedge \phi_2}$ is illustrated in Figure 2 (upper right). This policy guides the robot to first reach $a$ (except for state $s = 4$) and then go to $b$ which agrees with the specification.

Looking at Figure 1 , the FSA of $\phi = \phi_1 \wedge \phi_2$ have two options policies $\pi_\phi(\cdot, q_0)$ and $\pi_\phi(\cdot, q_1)$ [2] $(trap$ state and $q_f$ are terminal states which don't have options). State $q_1$ has only one outgoing edge with the guarding predicate $\psi_{q_1,q_f} : b$, which means $\pi_\phi(\cdot, q_1) = \pi_{\phi_1}(\cdot, q_2)$(they have the same guarding predicate). Policy $\pi_\phi(\cdot, q_0)$ is a switching policy between $\pi_{\phi_1}(\cdot, q_0)$ and $\pi_{\phi_2}(\cdot, q_0)$. Figure 2 (lower

---

[2]$\pi_\phi(\cdot, q_i)$ is the options policy of $\pi_\phi$ at automata state $q_i$ (definiton in Equation (9)). Writing in this form prevents cluttered subscripts

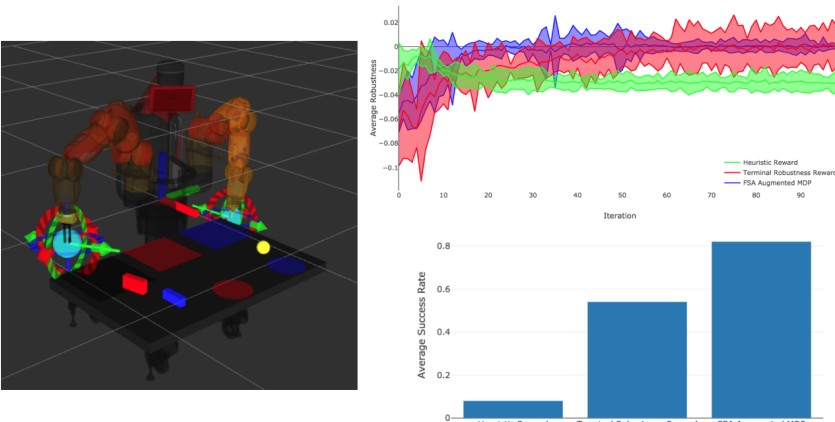

**Figure 3 :** (*left*): Baxter simulation Environment with three square regions (black,red, blue), two circular regions (red, blue), two boxes (red, blue) that the robot can manipulate and an interactive ball that the user can place anywhere on the table. Tasks are specified using these elements in Appendix A. (*upper right*): Learning curve for task $\phi_1$ over 5 random seeds. (*lower right*): Policy deployment success rate

left) shows the robustness comparison at each state. The policy with lower robustness is chosen following Equation (15). We can see that the robustness of both policies are the same from $s = -5$ to $s = 0$. And their policies agree in this range (Figures 3 and 4 ). As $s$ becomes larger, disagreement emerge because $\pi_{\phi_1}(\cdot, q_0)$ wants to stay closer to $b$ but $\pi_{\phi_2}(\cdot, q_0)$ wants otherwise. To maximize the robustness of their conjunction, the decisions of $\pi_{\phi_2}(\cdot, q_0)$ are chosen for states $s > 0$.

## 6.2 SIMULATED BAXTER

In this section, we construct a set of more complicated tasks that require temporal reasoning and evaluate the proposed techniques on a simulated Baxter robot. The environment is shown in Figure 3 (*left*). In front of the robot are three square regions and two circular regions. An object with planar coordinates $p = (x, y)$ can use predicates $\mathcal{S}_{red}(p), \mathcal{S}_{blue}(p), \mathcal{S}_{black}(p), \mathcal{C}_{red}(p), \mathcal{C}_{blue}(p)$ to evaluate whether or not it is within the each region. The predicates are defined by $\mathcal{S} : (x_{min} < x < x_{max}) \wedge (y_{min} < y < y_{max})$ and $\mathcal{C} : dist((x, y), (x, y)_{center}) < r$. $(x_{min}, y_{min})$ and $(x_{max}, y_{max})$ are the boundary coordinates of the square region, $(x, y)_{center}$ and $r$ are the center and radius of the circular region. There are also two boxes which planar positions are denoted as $p_{redbox} = (x, y)_{redbox}$ and $p_{bluebox} = (x, y)_{bluebox}$. And lastly there is an interactive ball that a user can move in space which 2D coordinate is denoted as $p_{sphere} = (x, y)_{sphere}$ (all objects move in the table plane).

We design seven tasks each specified by a scTLTL formula. The task specifications and their English translations are provided in Appendix A. Throughout the experiments in this section, we use proximal policy search Schulman et al. (2017) as the policy optimization method. The hyperparameters are kept fixed across the experiments and are listed in Appendix B. The policy is a Gaussian distribution parameterized by a feed-forward neural network with 2 hidden layers, each layer has 64 relu units. The state and action spaces vary across tasks and comparison cases, and are described in Appendix C.

We use the first task $\phi_1$ to evaluate the learning outcome using the FSA augmented MDP. As comparisons, we design two other rewards structures. The first is to use the robustness $\rho(s_{0:T}, \phi)$ as the terminal reward for each episode and zero everywhere else, the second is a heuristic reward that aims to align with $\phi_1$. The heuristic reward consists of a state that keeps track of whether the sphere is in a region and a set of quadratic distance functions. For $\phi_1$, the heuristic reward is

$$r_{\phi_1} = \begin{cases} -dist(p_{redbox}, p_{redsquarecenter}) & p_{sphere} \text{ is in red circle} \\ -dist(p_{redbox}, P_{\text{black square center}}) & \text{otherwise.} \end{cases} \quad (16)$$

Heuristic rewards for other tasks are defined in a similar manner and are not presented explicitly.

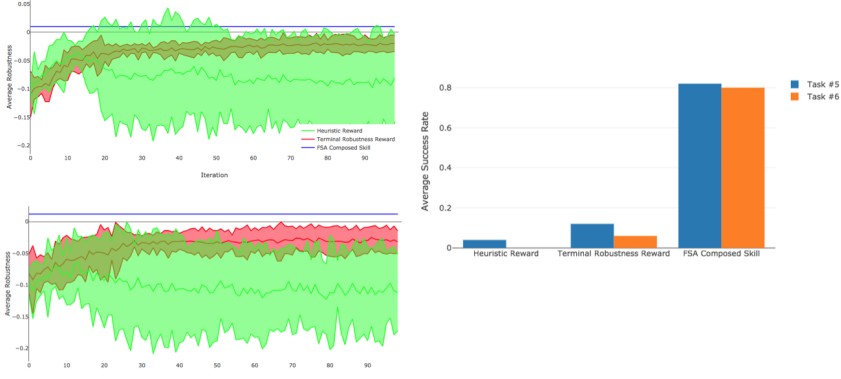

**Figure 4 :** (*left*): Learning curves for tasks $\phi_6$ and $\phi_7$ (task definitions provided in Appendix A). (*right*): Policy deployment success rate for tasks $\phi_6$ and $\phi_7$

The results are illustrated in Figure 3 (*right*). The upper right plot shows the average robustness over training iterations. Robustness is chosen as the comparison metric for its semantic rigor (robustness greater than zero satisfies the task specification). The reported values are averaged over 60 episodes and the plot shows the mean and 2 standard deviations over 5 random seeds. From the plot we can observe that the FSA augmented MDP and the terminal robustness reward performed comparatively in terms of convergence rate, whereas the heuristic reward fails to learn the task. The FSA augmented MDP also learns a policy with lower variance in final performance.

We deploy the learned policy on the robot in simulation and record the task success rate. For each of the three cases, we deploy the 5 policies learned from 5 random seeds on the robot and perform 10 sets of tests with randomly initialized states resulting in 50 test trials for each case. The average success rate is presented in Figure 3 (*lower right*). From the results we can see that the FSA augmented MDP is able to achieve the highest rate of success and this advantage over the robustness reward is due to the low variance of its final policy.

To evaluate the policy switching technique for skill composition, we first learn four relatively simple policies $\pi_{\phi_2}, \pi_{\phi_3}, \pi_{\phi_4}, \pi_{\phi_5}$ using the FSA augmented MDP. Then we construct $\pi_{\phi_6} = \pi_{\phi_2 \wedge \phi_3}$ and $\pi_{\phi_7} = \pi_{\phi_2 \wedge \phi_3 \wedge \phi_4 \wedge \phi_4}$ using Equation (15) (It is worth mentioning that the policies learned by the robustness and heuristic rewards do not have an automaton state in them, therefore the skill composition technique does not apply). We deploy $\pi_{\phi_6}$ and $\pi_{\phi_7}$ on tasks 6 and 7 for 10 trials and record the average robustness of the resulting trajectories. As comparisons, we also learn tasks 6 and 7 from scratch using terminal robustness rewards and heuristic rewards, the results are presented in Figure 4 . We can observe from the plots that as the complexity of the tasks increase, using the robustness and heuristic rewards fail to learn a policy that satisfies the specifications while the constructed policy can reliably achieve a robustness of greater than zero. We perform the same deployment test as previously described and looking at Figure 4 (*right*) we can see that for both tasks 6 and 7, only the policies constructed by skill composition are able to consistently complete the tasks.

## 7    CONCLUSION

In this paper, we proposed the FSA augmented MDP, a product MDP that enables effective learning of hierarchical policies using any RL algorithm for tasks specified by scTLTL. We also introduced automata guided skill composition, a technique that combines existing skills to create new skills without additional learning. We show in robotic simulations that using the proposed methods we enable simple policies to perform logically complex tasks.

Limitations of the current framework include discontinuity at the point of switching (for Equation (15)), which makes this method suitable for high level decision tasks but not for low level control tasks. The technique only compares robustness at the current step and chooses to follow a sub-policy for one time-step, making the switching policy short-sighted and may miss long term opportunities. One way to address this is to impose a termination condition for following each sub-

policy and terminate only when the condition is triggered (as in the original options framework). This termination condition can be hand designed or learned

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

# Appendix

## A   TASK SPECIFICATIONS

| Task | scTLTL Formula | English Description |
|------|----------------|---------------------|
| $\phi_1$ | $\big(\mathcal{C}_{red}(p_{ball}) \rightarrow \Diamond \mathcal{S}_{red}(p_{redbox})\big) \wedge$ $\big(\neg\mathcal{C}_{red}(p_{ball}) \rightarrow \Diamond \mathcal{S}_{black}(p_{redbox})\big)$ | If ball in red circle, then red box eventually in red square. Otherwise red box eventually in black square |
| $\phi_2$ | $\big(\mathcal{C}_{red}(p_{ball}) \rightarrow \Diamond \mathcal{S}_{red}(p_{redbox})\big) \wedge$ $\big(\neg(\mathcal{C}_{red}(p_{ball}) \vee (\mathcal{S}_{back}(p_{ball})) \rightarrow \Diamond \mathcal{S}_{black}(p_{redbox})\big)$ | If ball in red circle, then red box eventually in red square. If ball is not in red circle or black square, then red box eventually in black square |
| $\phi_3$ | $\big(\mathcal{C}_{blue}(p_{ball}) \rightarrow \Diamond \mathcal{S}_{blue}(p_{bluebox})\big) \wedge$ $\big(\neg(\mathcal{C}_{red}(p_{ball}) \vee (\mathcal{S}_{back}(p_{ball})) \rightarrow \Diamond \mathcal{S}_{black}(p_{bluebox})\big)$ | If ball in blue circle, then blue box eventually in blue square. If red ball is not in blue circle or black square, then blue box eventually in black square |
| $\phi_4$ | $\mathcal{S}_{black}(p_{ball}) \rightarrow \Diamond \mathcal{S}_{blue}(p_{redbox})$ | If ball in black square, then eventually red box in blue square |
| $\phi_5$ | $\mathcal{S}_{black}(p_{ball}) \rightarrow \Diamond \mathcal{S}_{red}(p_{bluebox})$ | If ball in black square, then eventually blue box in red square |
| $\phi_6$ | $\phi_2 \wedge \phi_3$ | Conjunction of task 2 and 3 |
| $\phi_7$ | $\phi_2 \wedge \phi_3 \wedge \phi_4 \wedge \phi_5$ | Conjunction of tasks 2, 3, 4, 5 |

## B   HYPERPARAMETERS FOR PROXIMAL POLICY OPTIMIZATION

| Hyperparameter | Value |
|----------------|-------|
| Num. Hidden Layers | 2 |
| Num. Units per layer | 64 |
| Activation | Relu |
| Policy Learning Rate | 0.009 |
| Value Learning Rate | 0.009 |
| Discount | 0.99 |
| Batch Size | 60 |
| GAE parameter | 0.99 |
| Num. Iterations | 100 |
| Num. Epochs | 20 |
| Clipping Parameter $\epsilon$ | 0.2 |
| Horizon | 20 |

## C  STATE AND ACTION SPACES

For experiments with the simulated Baxter robot, we delegate low level control to motion planning packages and only learn high level decisions. Depending on the task, the states are the planar positions of objects (red box, blue box, ball) and the automata state. The actions are the target positions of the objects. We assume that the low level controller can take objects to the desired target position with minor uncertainty that will be dealt with by the learning agent. The table below shows state and action spaces used for each task. $s_{\mathcal{M}}$ and $a_{\mathcal{M}}$ denote the spaces for regular MDP (used for terminal robustness rewards and heuristic rewards). $s_{\tilde{\mathcal{M}}}$ and $a_{\tilde{\mathcal{M}}}$ denote the spaces for FSA augmented MDP. $q$ denotes the automata state.

| Task | State Space | Action Space |
|------|-------------|--------------|
| $\phi_1$ | $s_{\mathcal{M}} = (p_{ball}, p_{redbox})$ 
 $s_{\tilde{\mathcal{M}}} = (p_{ball}, p_{redbox}, q)$ | $a_{\mathcal{M}/\tilde{\mathcal{M}}} = (p_{redbox})_{target}$ |
| $\phi_2$ | $s_{\mathcal{M}} = (p_{ball}, p_{redbox})$ 
 $s_{\tilde{\mathcal{M}}} = (p_{ball}, p_{redbox}, q)$ | $a_{\mathcal{M}/\tilde{\mathcal{M}}} = (p_{redbox})_{target}$ |
| $\phi_3$ | $s_{\mathcal{M}} = (p_{ball}, p_{bluebox})$ 
 $s_{\tilde{\mathcal{M}}} = (p_{ball}, p_{bluebox}, q)$ | $a_{\mathcal{M}/\tilde{\mathcal{M}}} = (p_{bluebox})_{target}$ |
| $\phi_4$ | $s_{\mathcal{M}} = (p_{ball}, p_{redbox})$ 
 $s_{\tilde{\mathcal{M}}} = (p_{ball}, p_{redbox}, q)$ | $a_{\mathcal{M}/\tilde{\mathcal{M}}} = (p_{redbox})_{target}$ |
| $\phi_5$ | $s_{\mathcal{M}} = (p_{ball}, p_{bluebox})$ 
 $s_{\tilde{\mathcal{M}}} = (p_{ball}, p_{bluebox}, q)$ | $a_{\mathcal{M}/\tilde{\mathcal{M}}} = (p_{bluebox})_{target}$ |
| $\phi_6$ | $s_{\mathcal{M}} = (p_{ball}, p_{redbox}, p_{bluebox})$ 
 $s_{\tilde{\mathcal{M}}} = (p_{ball}, p_{redbox}, p_{bluebox}, q_{\phi_2}, q_{\phi_3})$ | $a_{\mathcal{M}/\tilde{\mathcal{M}}} = (p, d)_{target}$ |
| $\phi_7$ | $s_{\mathcal{M}} = (p_{ball}, p_{redbox}, p_{bluebox})$ 
 $s_{\tilde{\mathcal{M}}} = (p_{ball}, p_{redbox}, p_{bluebox}, q_{\phi_2}, q_{\phi_3}, q_{\phi_4}, q_{\phi_5})$ | $a_{\mathcal{M}/\tilde{\mathcal{M}}} = (p, d)_{target}$ |

For tasks $\phi_6$ and $\phi_7$, the action space is three dimensional, the first two dimension $p = (x, y)$ is a target position, the third dimension $d$ controls which object should be placed at $p$. If $d < 0.5$, then $p = p_{redbox}$ and if $d > 0.5$, then $p = p_{bluebox}$.

# D   LEARNING WITH FSA AUGMENTED MDP (OFF-POLICY VERSION)

---

**Algorithm 1** Automata Guided RL (off-policy version)

---

1: **Inputs**: Episode horizon $T$, $\tilde{M}_\phi$ (consisting of an MDP and FSA $\mathcal{A}_\phi$), maximum size for replay pool $N$
2: Initialize parameterized policy $\pi^\theta$                          $\triangleright \theta$ is the policy parameters
3: Initialize replay pool $\mathcal{B} \leftarrow \{\}$
4: **for** $n = 1$ to *number of training episodes* **do**
5:      Select initial state $\tilde{s}_0 = (s_0, q_0)$       $\triangleright s_0$ can be randomly selected, $q_0$ is the initial automaton state $q^0$
6:      **for** t =0 to T **do** $a_t = \pi(\tilde{s}_t)$
7:          $\tilde{s}_{t+1} = $ GetNextState$(\tilde{s}_t, a_t)$
8:          **if** $q_{t+1} == q_f$ **then**
9:              $\tilde{r}_t = 2$                              $\triangleright$ terminal reward for satisfying $\phi$
10:              **break**                              $\triangleright \phi$ is satisfied, restart episode
11:          **else if** $q_{t+1} == trap$ **then**
12:              $\tilde{r}_t = -2$                              $\triangleright$ terminal reward for violating $\phi$
13:              **break**                              $\triangleright \phi$ is violated, restart episode
14:          **else**
15:              $\tilde{r}_t = $ GetReward$(\tilde{s}_t, \tilde{s}_{t+1})$                              $\triangleright$ using Equation 7
16:
17:          **end if**
18:          $\mathcal{B} \leftarrow (\tilde{s}_t, a_t, \tilde{s}_{t+1}, \tilde{r}_t)$                              $\triangleright$ store experience in replay pool
19:          **if** $size(\mathcal{B}) > N$ **then**
20:              pop$(\mathcal{B}[0])$
21:          **end if**
22:          $\theta \leftarrow $ UpdatePolicy$(\mathcal{B})$  $\triangleright$ this can be any RL update rule and doesn't necessarily have to occur at this location
23:      **end for**
24: **end for**

---

# E   SEMANTICS FOR SCTLTL

Following the syntax for scTLTL provided in Section 2.2, here we define the semantics for the language. We denote $s_t \in S$ to be the state at time $t$, and $s_{t:t+k}$ to be a sequence of states (state trajectory) from time $t$ to $t + k$, i.e., $s_{t:t+k} = s_t s_{t+1}...s_{t+k}$. The Boolean semantics of scTLTL is defined as:

$$
\begin{aligned}
s_{t:t+k} &\models f(s) < c &\Leftrightarrow\quad& f(s_t) < c, \\
s_{t:t+k} &\models \neg\phi &\Leftrightarrow\quad& \neg(s_{t:t+k} \models \phi), \\
s_{t:t+k} &\models \phi \Rightarrow \psi &\Leftrightarrow\quad& (s_{t:t+k} \models \phi) \Rightarrow (s_{t:t+k} \models \psi), \\
s_{t:t+k} &\models \phi \wedge \psi &\Leftrightarrow\quad& (s_{t:t+k} \models \phi) \wedge (s_{t:t+k} \models \psi), \\
s_{t:t+k} &\models \phi \vee \psi &\Leftrightarrow\quad& (s_{t:t+k} \models \phi) \vee (s_{t:t+k} \models \psi), \\
s_{t:t+k} &\models \bigcirc\phi &\Leftrightarrow\quad& (s_{t+1:t+k} \models \phi) \wedge (k > 0), \\
s_{t:t+k} &\models \Diamond\phi &\Leftrightarrow\quad& \exists t' \in [t, t+k)\ s_{t':t+k} \models \phi, \\
s_{t:t+k} &\models \phi\, \mathcal{U}\, \psi &\Leftrightarrow\quad& \exists t' \in [t, t+k)\ s.t.\ s_{t':t+k} \models \psi \\
&&& \wedge (\forall t'' \in [t, t')\ s_{t'':t'} \models \phi), \\
s_{t:t+k} &\models \phi\, \mathcal{T}\, \psi &\Leftrightarrow\quad& \exists t' \in [t, t+k)\ s.t.\ s_{t':t+k} \models \psi \\
&&& \wedge (\exists t'' \in [t, t')\ s_{t'':t'} \models \phi).
\end{aligned}
$$

A trajectory $s$ of horizon $T$ is said to satisfy formula $\phi$ if $s_{0:T} \models \phi$.

We also define the quantitative semantics for scTLTL (robustness degree), i.e., a real-valued function $\rho(s_{t:t+k}, \phi)$ of state trajectory $s_{t:t+k}$ and a scTLTL specification $\phi$ that indicates how far $s_{t:t+k}$ is from satisfying or violating the specification $\phi$. The quantitative semantics of scTLTL is defined as follows:

$$
\begin{aligned}
\rho(s_{t:t+k}, \top) &= \rho_{max}, \\
\rho(s_{t:t+k}, f(s_t) < c) &= c - f(s_t), \\
\rho(s_{t:t+k}, \neg\phi) &= -\rho(s_{t:t+k}, \phi), \\
\rho(s_{t:t+k}, \phi \Rightarrow \psi) &= \max(-\rho(s_{t:t+k}, \phi), \rho(s_{t:t+k}, \psi)) \\
\rho(s_{t:t+k}, \phi_1 \wedge \phi_2) &= \min(\rho(s_{t:t+k}, \phi_1), \rho(s_{t:t+k}, \phi_2)), \\
\rho(s_{t:t+k}, \phi_1 \vee \phi_2) &= \max(\rho(s_{t:t+k}, \phi_1), \rho(s_{t:t+k}, \phi_2)), \\
\rho(s_{t:t+k}, \bigcirc\phi) &= \rho(s_{t+1:t+k}, \phi)\ (k > 0), \\
\rho(s_{t:t+k}, \Diamond\phi) &= \max_{t' \in [t,t+k)} (\rho(s_{t':t+k}, \phi)), \\
\rho(s_{t:t+k}, \phi\, \mathcal{U}\, \psi) &= \max_{t' \in [t,t+k)} (\min(\rho(s_{t':t+k}, \psi), \\
&\qquad \min_{t'' \in [t,t')} \rho(s_{t'':t'}, \phi))), \\
\rho(s_{t:t+k}, \phi\, \mathcal{T}\, \psi) &= \max_{t' \in [t,t+k)} (\min(\rho(s_{t':t+k}, \psi), \\
&\qquad \max_{t'' \in [t,t')} \rho(s_{t'':t'}, \phi))),
\end{aligned}
$$

where $\rho_{max}$ represents the maximum robustness value. Moreover, $\rho(s_{t:t+k}, \phi) > 0 \Rightarrow s_{t:t+k} \models \phi$ and $\rho(s_{t:t+k}, \phi) < 0 \Rightarrow s_{t:t+k} \not\models \phi$, which implies that the robustness degree can substitute Boolean semantics in order to enforce the specification $\phi$ (refer to Li et al. (2016) for a more detailed description of TLTL and robustness).

