# OpenReview forum: "AUTOMATA GUIDED HIERARCHICAL REINFORCEMENT LEARNING FOR ZERO-SHOT SKILL COMPOSITION"
_ICLR.cc/2018/Conference — Reject_

### Official Review · AnonReviewer2 · 2017-11-26
**Makes a connection between LTL task representation and RL subtasks, providing some capability for composing skills.**

**Rating:** 5
**Confidence:** 4

**Review:**

The paper argues for structured task representations (in TLTL) and shows how these representations can be used to reuse learned subtasks to decrease learning time.

Overall, the paper is sloppily put together, so it's a little difficult to assess the completeness of the ideas. The problem being solved is not literally the problem of decreasing the amount of data needed to learn tasks, but a reformulation of the problem that makes it unnecessary to relearn subtasks. That's a good idea, but problem reformulation is always hard to justify without returning to a higher level of abstraction to justify that there's a deeper problem that remains unchanged. The paper doesn't do a great job of making that connection.

The idea of using task decomposition to create intrinsic rewards seems really interesting, but does not appear to be explored in any depth. Are there theorems to be had? Is there a connection to subtasks rewards in earlier HRL papers?

The lack of completeness (definitions of tasks and robustness) also makes the paper less impactful than it could be.

Detailed comments:

"learn hierarchical policies" -> "learns hierarchical policies"?

"n games Mnih et al. (2015)Silver et al. (2016),": The citations are a mess. Please proof read.

"and is hardly reusable" -> "and are hardly reusable".

"Skill composition is the idea of constructing new skills with existing skills (" -> "Skill composition is the idea of constructing
new skills out of existing skills (".

"to synthesis" -> "to synthesize".

"set of skills are" -> "set of skills is".

"automatons" -> "automata".

"with low-level controllers can" -> "with low-level controllers that can".

"the options policy π o is followed until β(s) > threshold": I don't think that's how options were originally defined... beta is generally defined as a termination probability.

"The translation from TLTL formula FSA to" -> "The translation from TLTL formula to FSA"?

"four automaton states Qφ = {q0, qf , trap}": Is it three or four?

"learn a policy that satisfy" -> "learn a policy that satisfies".

"HRL, We introduce the FSA augmented MDP" -> "HRL, we introduce the FSA augmented MDP.".

" multiple options policy separately" -> " multiple options policies separately"?

"Given flat policies πφ1 and πφ2 that satisfies " -> "Given flat policies πφ1 and πφ2 that satisfy ".

"s illustrated in Figure 3 ." -> "s illustrated in Figure 2 ."?

", we cam simply" -> ", we can simply".

"Figure 4 <newline> ." -> "Figure 4.".

", disagreement emerge" -> ", disagreements emerge"?

The paper needs to include SOME definition of robustness, even if it just informal. As it stands, it's not even clear if larger
values are better or worse. (It would seem that *more* robustness is better than less, but the text says that lower values are
chosen.)

"with 2 hidden layers each of 64 relu": Missing word? Or maybe a comma?

"to aligns with" -> "to align with".

" a set of quadratic distance function" -> " a set of quadratic distance functions".

"satisfies task the specification)" -> "satisfies the task specification)".

Figure 4: Tasks 6 and 7 should be defined in the text someplace.

"current frame work i" -> "current framework i".

" and choose to follow" -> " and chooses to follow".

" this makes" -> " making".

"each subpolicies" -> "each subpolicy".

---

> ### Author Response · Authors · 2018-01-03
> **Response to review2**
>
> Thank you for your detailed comments. We have incorporated all of them in our updated paper plus additional proofreading. The following are our attempts to answer your questions.
>
> 1. “. The problem being solved is not literally the problem of decreasing the amount of data needed to learn tasks, but a reformulation of the problem that makes it unnecessary to relearn subtasks. That's a good idea, but problem reformulation is always hard to justify without returning to a higher level of abstraction to justify that there's a deeper problem that remains unchanged.  ”
>
> We will try to address this concern but we are not certain that we’ve fully understood it. We provide solutions to two problems in this paper, the first is to use the FSA augmented MDP to impose hierarchical structure/constraint to the original MDP and by doing so enhance the sample efficiency and interpretability of policy learning using existing RL methods. The second is to take advantage of the structure of product automatons to compose new policy from existing policies. We didn’t intentionally try to reformulate the problem but rather to incorporate the extra knowledge and structure provided by temporal logic and FSA into the original problem and proposed methods to ensure that this incorporation is helpful to the learning process.
>
> 2. “The idea of using task decomposition to create intrinsic rewards seems really interesting, but does not appear to be explored in any depth. Are there theorems to be had? Is there a connection to subtasks rewards in earlier HRL papers?”
>
> The reasoning behind the construction of the intrinsic reward ($D^1_\phi$ in Definition 3) is to encourage the system to exit the current automaton state and eventually reach the final acceptance state which satisfies the TLTL specification or result in the trap state which given a large terminal penalty restarts the episode. This is a property of the FSA and therefore we didn’t go into much depth (we added a reference in section 2.2 (Vasile 2017) that show the application of similar idea).
>
> As far as we know, existing HRL methods require some kind of human effort to engineer the hierarchical structure into the learner. This ranges from explicitly defining the options (initial set, policy and terminal condition) in the original options paper to designing the finite state machine in HAM (hierarchies of abstract machines). More recent efforts have relaxed these requirements of defining what each option is and how they interact with each other and depend on the learning algorithm to figure out the specifics, however the user still has to define either the intrinsic motivation (such as the h-DQN by Kulkarni et al) or the number of options to discover (such as the option-critic framework by Bacon et al). Our work free the user from designing any of the above and utilize the FSA to provide hierarchical structure and motivation. Due to space constraints, a detailed literature review on HRL is not included.
>
> 3. “The lack of completeness (definitions of tasks and robustness) also makes the paper less impactful than it could be.“
>
> We originally included a reference to the paper containing the definition of robustness, but given that this has caused enough confusion, we have provided the full definition of the boolean and quantitative semantics along with small examples in Appendix E. We are not sure what exactly “definitions of tasks” stands for but we used TLTL formula as task specifications and examples of them are provided in Appendix A.
>
> 4. “The paper needs to include SOME definition of robustness, even if it just informal. As it stands, it's not even clear if larger values are better or worse. (It would seem that *more* robustness is better than less, but the text says that lower values are chosen.)”
>
> The higher the robustness the better satisfaction of the specification. The reason Equation (15) chooses the policy with lower robustness is because we are trying to maximize the minimum of two robustnesses (Equation (13)) with means we have to maximize the lower of the two, assuming that following each policy maximizes its own robustness at the step level (which is a limitation of the current method and discussed in the conclusion, we are working on improving this).

---

### Official Review · AnonReviewer1 · 2017-11-27
**Good objective but I don't think the primary result is correct**

**Rating:** 3
**Confidence:** 4

**Review:**

I very much appreciate the objectives of this paper:  learning compositional structures is critical for scaling and transfer.

The first part of the paper offers a strategy for constructing a product MDP out of an original MDP and the automaton associated with an LTL formula, and reminds us that we can learn within that restricted MDP.  Some previous work is cited, but I would point the authors to much older work of Parr and Russell on HAMs (hierarchies of abstract machines) and later work by Andre and Russell, which did something very similar (though, indeed, not in hybrid domains).  The idea of extracting policies corresponding to individual automaton states and making them into options seems novel, but it would be important to argue that those options are likely to be useful again under some task distribution.

The second part offers an exciting result:  If we learn policy pi_1 to satisfy objective phi_1 and policy pi_2 to satisfy objective phi_2, then it will be possible to switch between pi_1 and pi_2 in a way that satisfies phi_1 ^ phi_2.   This just doesn't make sense to me.  What if phi_1 is o ((A v B) Until C) and phi_2 is o ((not A v B) Until C).   Let's assume that o(B Until C) is satisfiable, so the conjunction is satisfiable.  However, we may find policy pi_1 that makes A true and B false (in general, there is no single optimal policy) and find pi_2 that makes A false and B false, and it will not be possible to satisfy the phi_1 and phi_2 by switching between the policies.    But, perhaps I am misunderstanding something.

Some other smaller points:
- "zero-shot skill composition" sounds a lot like what used to be called "planning" or "reasoning"
- The function rho is originally defined on whole trajectories but in eq 7 it is only on a single s':  I'm not sure exactly what that means.
- Section 4:  How is "as soon as possible" encoded in this objective?
- How does the fixed horizon interact with conjoining goals?
- There are many small errors in syntax;  it would be best to have this paper carefully proofread.

---

> ### Author Response · Authors · 2018-01-03
> **Response to review1**
>
> Thank you for your comments. The following are our attempts to answer your questions.
>
> 1. “The idea of extracting policies corresponding to individual automaton states and making them into options seems novel, but it would be important to argue that those options are likely to be useful again under some task distribution”
>
> Each option that corresponds to an automaton state q satisfies the predicate defined by $D^q_\phi$ (in Definition 3). Since the FSA for an LTL formula is constructed by the conjunction, disjunction, and negation of various predicates, the already learned options can be used as is or to construct policies that satisfy new LTL specifications given the state and action distributions remain the same.
>
> 2. “Section 4:  How is "as soon as possible" encoded in this objective?”
>
> This is our neglect in proofreading, but a discount factor in addition to the terminal reward are used to ensure “as soon as possible”. We’ve made this correction in our updated version
>
>
> 3. “ What if phi_1 is o ((A v B) Until C) and phi_2 is o ((not A v B) Until C).   Let's assume that o(B Until C) is satisfiable, so the conjunction is satisfiable.  However, we may find policy pi_1 that makes A true and B false (in general, there is no single optimal policy) and find pi_2 that makes A false and B false, and it will not be possible to satisfy the phi_1 and phi_2 by switching between the policies. ”
>
>
> This is a good example and we’ll try our best to clarify. Given the learning objective in Equation (7), if the initial condition doesn’t violate (A v B), then $pi_1$ will head for C while ensuring that (A v B) is satisfied along its path to making C true. A transition from A=true to B=true is possible if the shortest path to C requires so (shortest path is the result of the discount factor and terminal reward). If the initial position violates (A v B), then the episode restarts (results in the trap state). The same goes for $pi_2$. So for both $phi_1$ and $phi_2$, the goal is to quickly get to C while satisfying (A v B) and (not A v B) respectively. Under Definition 3, the optimal policies for $pi_1$ and $pi_2$ are unique. The necessary condition for $phi_1 ^ phi_2$ to be satisfiable is that the intersection of B and C is nonempty and the initial condition satisfies (A v B) ^ (not A v B) = B v (A ^ not A) = B. Having met these conditions, choosing either $pi_1$ or $pi_2$ will result in satisfaction of $phi_1 ^ phi_2$
>
> 4. "zero-shot skill composition" sounds a lot like what used to be called "planning" or "reasoning"
>
> We understand “planning” and “reasoning” as obtaining the optimal policy under known system transition function.
>
> 5. “The function rho is originally defined on whole trajectories but in eq 7 it is only on a single s':  I'm not sure exactly what that means.”
>
> Thank you for raising this confusion, we’ve added a footnote to Equation (7) as well as the full definition for robustness in Appendix E
>
> 6. “How does the fixed horizon interact with conjoining goals?”
>
> We’re not sure what “conjoining goals” means.
>
> 7. “There are many small errors in syntax;  it would be best to have this paper carefully proofread.”
>
> We’ve put much effort in proofreading and have uploaded the newer version

---

### Official Review · AnonReviewer3 · 2017-11-28
**This paper presents a method to connect truncated linear temporal logic formulas to reinforcement learning policies, but several relevant details are not sufficiently clear.**

**Rating:** 4
**Confidence:** 3

**Review:**

This paper proposes to join temporal logic with hierarchical reinforcement learning to simplify skill composition.  The combination of temporal logic formulas with reinforcement learning was developed previously in the literature, and the main contribution of this paper is for fast skill composition.  The system uses logic formulas in truncated linear temporal logic (TLTL), which lacks an Always operator and where the LTL formula (A until B) also means that B must eventually hold true. The temporal truncation also requires the use of a specialized MDP formulation with an explicit and fixed time horizon T.  The exact relationship between the logical formulas and the stochastic trajectories of the MDP is not described in detail here, but relies on a robustness metric, rho.  The main contributions of the paper are to provide a method that converts a TLTL formula that specifies a task into a reward function for a new augmented MDP (that can be used by a conventional RL algorithm to yield a policy), and a method for quickly combining two such formulas (and their policies) into a new policy.  The proposed method is evaluated on a small Markov chain and a simulated Baxter robot.

The main problem with this paper is that the connections between the TLTL formulas and the conventional RL objectives are not made sufficiently clear.  The robustness term rho is essential, but it is not defined.  I was also confused by the notation $D_\phi^q$, which was described but not defined.  The method for quickly combining known skills (the zero-shot skill composition in the title) is switching between the two policies based on rho.  The fact that there may be many policies which satisfy a particular reward function (or TLTL formula) is ignored.  This means that skill composition that is proposed in this paper might be quite far from the best policy that could be learned directly from a single conjunctive TLTL formula. It is unclear how this approach manages tradeoffs between objectives that are specified as a conjunction of TLTL goals. is it better to have a small probability of fulfilling all goals, or to prefer a high probability of fulfilling half the goals?  In short the learning objectives of the proposed composition algorithm are unclear after translation from TLTL formulas to rewards.

---

> ### Author Response · Authors · 2018-01-03
> **Response to review3**
>
> Thank you for your comments, the following are our attempts to address your questions and concerns.
>
> 1. “The combination of temporal logic formulas with reinforcement learning was developed previously in the literature, and the main contribution of this paper is for fast skill composition” and “The main contributions of the paper are to provide a method that converts a TLTL formula that specifies a task into a reward function for a new augmented MDP ”
>
> Compared to similar ideas in previous literature, we extended the combination of temporal logic and RL to hybrid domains and proposed the FSA augmented MDP as a bridge between the learned flat policy and the hierarchical structure of the task. By doing so, options can be easily learned and extracted from the flat policy without the need manually design the specifics of the hierarchy. The FSA that results from the TLTL formula does not only provide the extrinsic and intrinsic rewards but also the temporal constraints of how the task should proceed which is incorporated into the system dynamics in Equation (6).
>
> 2. “The exact relationship between the logical formulas and the stochastic trajectories of the MDP is not described in detail here, but relies on a robustness metric, rho” and “ The robustness term rho is essential, but it is not defined”
>
> Thank you for raising this concern, we’ve originally made a reference to the paper containing the definition of robustness in Section 2.1, but now we have also added the full definition in Appendix E.
>
> 3. “The main problem with this paper is that the connections between the TLTL formulas and the conventional RL objectives are not made sufficiently clear.”
>
> We try to make this connection in Definition 1 and Problem 1, specifically Equation (1) and Equation (5). The goal of conventional RL is to maximize the expected return, the goal of RL with TLTL specification is to maximize the expected satisfaction of the TLTL formula.
>
> 4. “I was also confused by the notation $D_\phi^q$, which was described but not defined”
>
> $D_\phi^q$ is defined in Definition 3 in the text after Equation (7). An example is provided after Equation (8).
>
> 5. “The fact that there may be many policies which satisfy a particular reward function (or TLTL formula) is ignored.  This means that skill composition that is proposed in this paper might be quite far from the best policy that could be learned directly from a single conjunctive TLTL formula.”
>
> The optimal policy for the FSA augmented MDP is unique under the effect of a discount factor and the terminal reward (we carelessly neglected the discount factor during proofreading which is now added). The optimal policy should guide the system out of the current automaton state as fast as possible and towards the final accepting state. Therefore, given enough terminal motivation, the desired behavior is to find the shortest path to satisfying the specification at any given state. And the composed policy will also achieve this following the characteristics of the product automaton and the derivations in Section 5. However, if we assume no discount factor (discount=1), we end up with a set of (possibly infinite) satisfying policies. The composed policy will thus also be one of many satisfying policies that satisfies the conjunction of two TLTL specs. Depending on how hyperparameters are set, the composed policy is likely different from that learned directly from a single conjunctive TLTL formula, but their expected return will be the same (given the terminal rewards are set up to encourage the same behavior). Optimality aside, the goal of finding a satisfying policy given by Problem 1 and Problem 2 will be met.
>
> 6. “It is unclear how this approach manages tradeoffs between objectives that are specified as a conjunction of TLTL goals. is it better to have a small probability of fulfilling all goals, or to prefer a high probability of fulfilling half the goals?  In short the learning objectives of the proposed composition algorithm are unclear after translation from TLTL formulas to rewards.”
>
> For the skill composition part, the objective is to fulfill all goals and hence the conjunction. If we only want to fulfill a subset of all the goals, then a disjunction would be used and the policy switching scheme would be slightly different but easily adaptable (part of our on-going work). There is not the notion of probability and we hope to show that by using our method it can be guaranteed that the conjunctive goal is fulfilled given that each sub-policy can fulfill their own goal.

---

### Decision · Program_Chairs · 2018-01-29
**ICLR 2018 Conference Acceptance Decision**

**Decision:**

Reject

**Comment:**

The authors make an argument for constructing an MDP from the formal structures of temporal logic and associated finite state automata and then applying RL to learn a policy for the MDP. This does not provide a solution for low-level skill composition, because there are discontinuities between states, but does provide a means for high level skill composition.

The reviewers agreed that the paper suffered from sloppy writing and unclear methods. They had concerns about correctness, and were not impressed by the novelty (combining TL and RL has been done previously). These concerns tip this paper to rejection.